# Comparative analysis of atmospheric radiative transfer models using the Atmospheric Look-up table Generator (ALG) toolbox (version 2.0)

Jorge Vicent[1,2], Jochem Verrelst[2], Neus Sabater[3], Luis Alonso[2], Juan Pablo Rivera-Caicedo[4], Luca Martino[5], Jordi Muñoz-Marí[2], and José Moreno[2]

[1]Magellium, Toulouse, France. Web: http://www.magellium.com.
[2]Image Processing Laboratory, Universitat de València, 46980 Paterna (Valencia), Spain.
[3]Finnish Meteorological Institute, Erik Palménin aukio 1, FI-00560 Helsinki (Finland).
[4]Secretary of Research and Graduate Studies, CONACYT-UAN, 63155 Tepic, Nayarit, Mexico
[5]Departamento de Teoría de la Señal y Comunicaciones, Universidad Rey Juan Carlos, 28943 Fuenlabrada (Madrid), Spain.

**Correspondence:** Jorge Vicent (jorge.vicent@uv.es)

**Abstract.** Atmospheric radiative transfer models (RTMs) are software tools that help researchers in understanding the radiative processes occurring in the Earth's atmosphere. Given their importance in remote sensing applications, the intercomparison of atmospheric RTMs is therefore one of the main tasks to evaluate model performance and identify the characteristics that differ between models. This can be a tedious tasks that requires a good knowledge of the model inputs-outputs and generation of large databases of consistent simulations. With the evolution of these software tools, their increase in complexity bears implications towards their use in practical applications and model intercomparison. Existing RTM-specific graphical user interfaces are not optimized for performing intercomparison studies of a wide variety of atmospheric RTMs. In this paper, we present the *Atmospheric Look-up table Generator* (ALG) version 2.0, a new software tool that facilitates generating large databases for a variety of atmospheric RTMs. ALG facilitates consistent and intuitive user interaction to enable running model executions and storing RTM data for any spectral configuration in the optical domain. We demonstrate the utility of ALG to perform intercomparison studies of radiance simulations from broadly used atmospheric RTMs (6SV, MODTRAN, libRadtran) through global sensitivity analysis. We expect that providing ALG to the research community will facilitate the usage of atmospheric RTMs to a wide range of applications in Earth Observation.

## 1 Introduction

Atmospheric radiative transfer models (RTMs) have deeply helped in understanding the radiation processes occurring in the Earth's atmosphere (Dubovik and King, 2000; Iacono et al., 2008). RTMs are physically-based computer models that numerically describe the absorption, emission and scattering processes in the ultraviolet to microwave region. Therefore, they are widely used in Earth observation scientific and technological applications, such as: (i) sensor/mission design (Kerekes et al., 1999; Verhoef and Bach, 2012; Verstraete et al., 2015), (ii) atmospheric chemistry (Theys et al., 2007; Dubovik et al., 2011), (iii) meteorology and climatology (Forster et al., 2011), (iv) atmospheric correction (Richter, 1996; Cooley et al., 2002; North

et al., 2008), and (v) atmospheric physics (Stamnes et al., 1988). Over time and through continuous improvements, these models have increased in realism from simple semi-parametric equations (e.g., Seidel et al., 2010) towards advanced RTMs that allow for explicit 3D representations of complex interactions in the atmosphere. Some examples include: 6SV (Vermote et al., 1997), libRadtran (Mayer and Kylling, 2005; Emde et al., 2016), MODTRAN (Berk et al., 2006, 2014), MOMO (Fell and Fischer, 2001) and RTTOV (Saunders et al., 2018).

Given the importance of atmospheric RTMs for remote sensing applications, their intercomparison is one of the main tasks in order to determine their performance and identify the characteristics that differ between models (Kotchenova et al., 2008; Seidel et al., 2010; Proud et al., 2010; Callieco and Dell'Acqua, 2011). The process of comparing various atmospheric RTMs can be a tedious task that requires a good knowledge of the model inputs-outputs and the generation of large database of consistent simulations. Indeed, the evolution of RTMs towards more advanced models has resulted in an increase in complexity and intepretability of these models, which bears implications towards practical implementation of intercomparison studies. To overcome this limitation, Graphical User Interfaces (GUI) have been developed to facilitate RTM use and execution. A few examples of these GUIs can be found for 6SV (Matarrese et al., 2015; Wilson, 2013), MODTRAN (Schläpfer, 2016; Berk et al., 2017) or libRadtran (Mayer and Kylling, 2017). These well-documented tools allow complete access to all functionalities and configuration parameters of the models they were designed for, including user-support and continuous updates. However, each of these GUIs are customized for their specific RTM; and none can be used to define and run simulations for multiple RTMs in a consistent manner. In addition, they are not designed to easily precompute large databases, which are important due to the high computational burden for performing statistical analysis (Verrelst et al., 2016) or running these models in a pixel-per-pixel basis (Gastellu-Etchegorry et al., 2003; Guanter et al., 2009). Altogether, these GUIs are not fully offering practical solutions for the implementation of atmospheric RTMs in Earth Observation applications and, in particular, for model intercomparison. Users of atmospheric RTMs are therefore obliged to develop their own specific scripts to create datasets, which are typically: (1) limited to a handful input variables and (2) hardly extensible to other RTMs.

In an attempt to facilitate the consistent simulation of databases for a wide range of atmospheric RTMs, we developed the *Atmospheric Look-up table Generator* (ALG). ALG is a Matlab-compiled software package that allows generating Look-Up Tables (LUT) based on a suite of atmospheric RTMs. Namely, a LUT consists of a collection of input atmospheric conditions and corresponding generated RTM spectral outputs (see Section 3.3 for further details). ALG provides consistent and intuitive user interaction for defining model configuration, running and storing RTM data for any spectral configuration in the optical domain. The main objectives of this paper are therefore: (1) to describe the ALG tool from a functional and sofware design perspective, thereby giving the reader an overview of the implemented features and generated LUT data; and (2) to perform a comparison study between the models implemented in ALG: MODTRAN (v5 and v6), 6SV v2.1 and libRadtran v2.0.2.

The remainder of this work is structured as follows: Section 2 gives an overview of the currently implemented atmospheric RTMs and associated graphical interfaces. Section 3 describes the ALG software design and its main features. Section 4 provides a comparative analysis of the implemented atmospheric RTMs. Section 5 summarizes a few applications as examples of the usage of ALG. Finally, Section 6 concludes with an outlook of on-going and planned functionalities to be implemented in future versions of ALG.

## 2 Overview of existing atmospheric RTMs and associated GUIs

In this section we briefly describe the key features of the atmospheric RTMs compatible with ALG version 2.0 and their associated user interfaces.

### 2.1 MODTRAN

Developed by Spectral Science Inc. (www.modtran.spectral.com), MODTRAN (Berk et al., 2006, 2014) is one of the most widely used RTMs by scientists and commercial organizations with multiple applications in Earth Observation. MODTRAN solves the atmospheric radiative transfer (RT) equation with the accurate Discrete Ordinates (DISORT) method (Stamnes et al., 1988) and a statistical simulation of the absorption effects through the correlated-k method (Goody et al., 1989). The coupled absorption and scattering simulations are calculated in a stratified spherically-symmetric atmosphere consituted of vertical

profiles of molecules (e.g., Anderson et al., 1986). Suspended particles are divided into the boundary layer aerosols (<2 km) and stratospheric aerosols. Accordingly, MODTRAN combines the effects of molecular and particulate absorption/emission and scattering, surface reflections and emission, solar/lunar illumination, and spherical refraction. The calculated spectral outputs include direct and diffuse transmittance, top-of-atmosphere (TOA) radiance fluxes, solar/lunar irradiance, horizontal fluxes, cooling rates, etc. These outputs extend from the ultraviolet to the long wavelength infrared spectral range (0.2-200 μm) and

are provided at a resolution up to 0.1 $cm^{-1}$ (0.001-0.1 nm in the VIS-SWIR spectral range) for the narrow band simulations, or even higher using the line-by-line capabilities (Berk and Hawes, 2017). With over 30 years of heritage, MODTRAN has been extensively validated, and it continues to be maintained and upgraded (Berk et al., 2015).

   Several GUIs are made available by commercial companies such as Spectral Sciences Inc., Ontar's PcModWin (www.ontar.com) and ReSe's MODO (Schläpfer, 2016). All these tools consist of a graphical front-end that wraps around MODTRAN, facilitat-

ing user interaction and model configuration from scratch and thus leveraging the use of MODTRAN. These GUIs give access to a wide range of input parameters such as definition of vertical profiles, geometric conditions and spectral configuration. Users can therefore format the input files to run MODTRAN and display the resulting simulations through interactive plotting panels. Some of these tools also allow running several simulations through the GUI, manually varying the configuration of every new simulation or through parameter series of one parameter at a time. Despite these capabilities, none of these tools are

customized to generate large LUTs of MODTRAN simulations.

### 2.2 6SV

6S was developed in the 90s  (Vermote et al., 1997). Since then, it has been applied to process broadband resolution instruments (e.g., El Hajj et al., 2008; Matthews et al., 2010; Liu et al., 2012). 6S solves the RT equation based on the method of successive orders of scattering (Lenoble, 1985), with a decoupling of the absorption and scattering effects by molecules and

particulates. These numerical approximations are performed in a stratified plane-parallel atmosphere consituted of vertical profiles of molecules and aerosols. An exponential vertical profile is used for the aerosol concentration and the optical properties are assumed to be the same in the entire atmospheric column. The calculated spectral outputs include direct and diffuse trans-

mittance in the sun-to-target and target-to-sensor directions, spherical albedo, atmospheric path radiance and TOA radiance fluxes. These outputs extend from the spectral range between 0.3-4 μm at a resolution of 2.5 nm. The latest updates of the code account for polarization in the atmosphere (Kotchenova et al., 2006; Kotchenova and Vermote, 2007).

The only GUI dedicated to 6S known by the authors is its official website (6s.ltdri.org). Under its section *Run 6SV*, users can define the input configuration and run the code to retrieve the 6S input and output files directly from the web browser. Accordingly, the generation of multi-parametric LUTs is not feasible with this online GUI. In order to overcome this limitation, Py6S was developed (Wilson, 2013). Py6S is a Python-based Application Programming Interface that provides (1) user-friendly model setting, (2) run and plotting capabilities, and (3) ability to import external data (e.g. atmospheric profiles). As such, Py6S can be integrated in any Python code facilitating the direct usage of 6S in data processing algorithms or for LUT generation.

## 2.3 LibRadtran

The libRadtran software package is a collection of algorithms for atmospheric radiative transfer calculations (www.libradtran.org) and thus used for various applications in the field of remote sensing, atmospheric physics and climatology. LibRadtran implements different solvers of the RT equation (DISORT among them) that allow computing (polarized) radiances, irradiances and actinic fluxes in the solar and thermal spectral regions with a resolution up to $1 \text{ cm}^{-1}$ (0.01-0.6 nm in the VIS-SWIR spectral range) (Mayer and Kylling, 2005; Emde et al., 2016). LibRadtran is a user-friendly RTM that, similar to MODTRAN, allows users to define to configure the atmospheric state with a wide variety of options, including molecules, aerosols water/ice clouds and surface boundary conditions. The most recent updates include new features such as: (1) simulation of the Raman scattering, (2) new parameterization of molecular absorption called Reptran (Gasteiger et al., 2014) and aerosol optical properties, or (3) Monte-Carlo solver of the RT equation. The flexible design of libRadtran makes it a powerful and versatile tool for research tasks. Furthermore, libRadtran includes a Python-based graphical user interface that simplifies the usage of the model. The GUI has similar functionalities as those previously discussed for MODTRAN. As such, it is not possible to run a large set of simulations and compile LUTs for later use in data processing applications.

## 2.4 OPAC

Despite of not being an atmospheric RTM per se, the OPAC package is a widely used software tool that provides aerosol optical properties in the 0.25 and 40 μm spectral range (Hess et al., 1998; Koepke et al., 2015). OPAC calculates the extinction, scattering, and absorption coefficients, the single scattering albedo, the asymmetry parameter, and the phase function. These optical properties are calculated for a set of 10 pre-defined aerosol models and user-defined mixtures, thus expanding the existing capabilities of atmospheric RTMs.

Similar to the previously defined RTMs, OPAC operates on the basis of input/output files. In order to facilitate its use, several GUIs have been developed that are compatible with OPAC. MOSPMAP is a toolbox, linked with libRadtran, for the optical modelling of complex aerosols, including pre-calculated optical properties of single aerosol particles as those in the OPAC package (Gasteiger and Wiegner, 2018). A user-friendly web interface was developed for MOPSMAP facilitating online calculations. The AEROgui tool (Pedrós et al., 2014) is a similar GUI package that can be used to obtain the optical properties

of a mixture of aerosol particles. Accordingly, AEROgui expands the current capabilities of OPAC by also providing a user interface to facilitate user definition of new aerosol mixtures. However, in most cases, there is no direct and straighforward way to include the OPAC output data into atmospheric RTM simulations.

## 3 The Atmospheric Look-up table Generator (ALG) tool

In this section we start identifying the key software functionalities (Sect. 3.1). Then we introduce the ALG graphical interface and how it is used to configure a new LUT (Sect. 3.2). Finally, we describe how ALG automatically generates a LUT and its content (Section 3.3).

### 3.1 Key software functionalities

The primary goal of ALG is to provide a scientific software package that fills the gaps observed in the previously analyzed tools.
In particular: (1) Each existing GUI is compatible with only one specific atmospheric RTM (e.g., PcModWin for MODTRAN) and cannot be used to configure and run simulations for other RTMs. (2) These tools are not intended to run a large number of simulations and thus creating LUTs. (3) The inputs and outputs of each atmospheric RTM are generally not consistent between each other, adding an extra layer of complexity when using or comparing various models.

Accordingly, ALG offers the following key functionalities:

1. ALG functions as a wrapper for running atmospheric RTMs, providing a graphical tool in which users can select the input configuration (i.e., atmospheric, geometric and spectral). In this way, ALG keeps the same functionality as all the previously described tools.

2. ALG facilitates the integration of additional atmospheric RTMs. In its current version 2.0, ALG is compatible with MODTRAN5 (Berk et al., 2006), MODTRAN6 (Berk et al., 2014), 6SV version 2.1 (Vermote et al., 1997) and libRadtran
version 2.0.2 (Emde et al., 2016).

3. The GUI is common to all the implemented models so that it facilitates the configuration of a wide variety of atmospheric RTM.

4. LUT design with ALG is a flexible process in which users can select a RTM, its input atmospheric variables and values.

5. ALG automatically processes and harmonizes all the RTM input and output data into the final LUT file. With this
functionality, ALG facilitates the intercomparability between atmospheric RTMs and the possibility to alternate between models in a data processing algorithm (e.g., for atmospheric correction).

6. ALG provides a help system and a set of tutorials to facilitate users with the installation and operation of the software.

## 3.2 ALG graphical interface

ALG's graphical interface provides users the tools to configure the software, to run the RTM simulations and to construct the final LUT. It is divided into three main elements: the *Software configuration*, the *LUT configuration* and the *Help system* GUIs.

The *Software configuration* GUI facilitates the user to edit software aspects of ALG such as: (1) the path to the executable RTM files, (2) the default folder to store the output data, and (3) the default CPU-cores used to run a RTM. In addition, users can add new RTM input variables and edit their default values. This software configuration GUI also permits editing and storing the spectral configuration of existing and user-defined remote sensing instruments to ease the generation of sensor-specific LUTs.

In its core interface (*LUT configuration*) users can select the RTM input variables and values used to run the simulations and to store the spectral outputs into the LUT. This GUI is based on the commonalities found in Section 2, with extended functionalities that allow running a large set of simulations. The LUT configuration GUI is divided in five main subsequent steps as shown in Fig. 1 and further described in the paragraphs below.

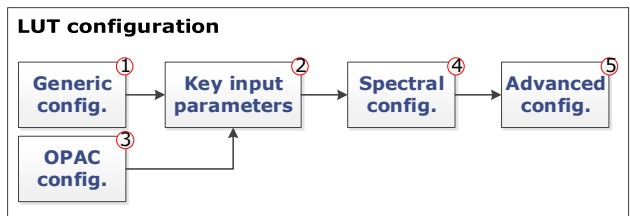

**Figure 1.** LUT configuration steps accesible through ALG's graphical interface.

In step 1 (*Generic configuration*), it is selected the atmospheric RTM used to run the simulation and the sampling method to distribute the LUT nodes (i.e., collection of points of input atmospheric and geometric variables). Several methods are implemented to distribute LUT nodes: (a) systematic gridded combinations of all input values, typically applied in atmospheric correction algorithms (e.g., Guanter et al., 2009); (b) scattered near-random and homogeneous sampling of the input variable space based on Latin Hypercube Sampling (McKay et al., 1979), Sobol distribution (Bratley and Fox, 1988) and Halton distribution (Kocis and Whiten, 1988); or (c) automatic gradient-based distribution (Vicent et al., 2018). Parallel instances of the selected atmospheric RTM are invoked in order to speed up the process of generating large LUTs (Brazile et al., 2008).

In step 2 (*Key input parameters*), ALG allows users to introduce selected atmospheric and geometric variables and their values (see Fig. 2). In ALG, input variables are divided into two types: discrete and continuous. Discrete variables are those that can only take on a certain number of values. Typical examples of discrete variables are the atmospheric profile, the aerosol model or the extraterrestrial solar irradiance. Continuous variables can have any value within an allowed range. Typical examples of continuous variables are the columnar water vapor (CWV), the aerosol optical thickness (AOT) or the solar/viewing zenith angle (SZA, VZA). For continuous variables, their values are varying between an user-input minium/maximum range and, in case of gridded sampling, distributed according to a selected distribution (linear, logarithmic, exponential or cosine).

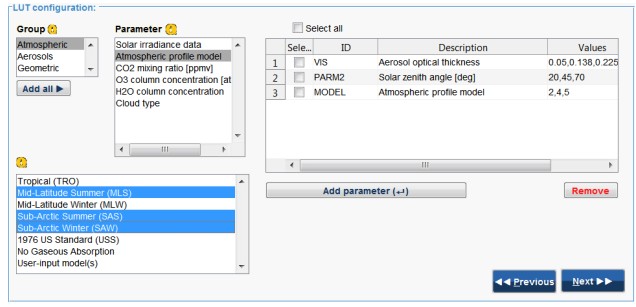

**Figure 2.** The *key input parameters* of the LUT configuration GUI (see step 2 in Fig. 1) allows users to introduce input model variables and their values.

In step 3 (*OPAC configuration*), ALG implements a back-end interface with OPAC (v3.1) database, expanding the predefined aerosol models with a comprehensive database of aerosol optical properties (i.e., extinction, absorption and phase function). For OPAC aerosol models, users can create new aerosol mixtures described by their particle number density from a set of basic components.

In step 4 (*Spectral configuration*), the spectral configuration of the RTM simulations is introduced. Users can set the desired spectral range and resolution, eventually at non-contiguous spectral intervals, saving computation time and disk storage of unwanted wavelengths. A set of predefined spectral configuration of common satellite instruments or user-defined sensors can be loaded.

Finally, in step 5 (*Advanced configuration*), the user has access to advanced RTM configuration parameters (e.g., selection of radiative transfer solver, printed output files). These parameters largely depend on the selected RTM.

All these LUT configuration parameters are stored in a `.xml` file that is later used by ALG's internal functions (see Section 3.3) to automatically run the RTM simulations and construct the final LUT. This configuration file can be loaded by ALG, allowing users to edit and re-run previous simulations e.g., by adding new atmospheric variables, changing the spectral configuration or modifying advanced settings. It worth also noticing that the LUT configuration interface is common for all implemented RTMs and the software harmonizes the naming and definition of atmospheric and geometric parameters to all models.

Additionally, ALG's GUI provides access to the help system with information about: (1) how to install the software and third-party RTMs, (2) how to generate a new LUT, (3) sample cases (tutorials) with practical applications of the use of the software, and (4) implemented RTMs and input variables. The ALG help system is based on Matlab® help browser developed by ©The MathWorks, Inc.

### 3.3 ALG internal functions. Look-up table generation

After setting the LUT configuration (see Section 3.2), ALG implements a set of back-end functionalities to automatically generate the output atmospheric LUT based on the input configuration (see Fig. 3).

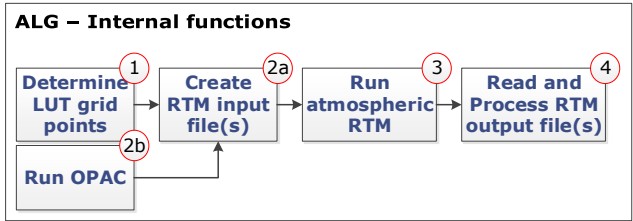

**Figure 3.** ALG's internal functions for RTM model execution and LUT generation process.

In step 1, ALG starts determining the LUT nodes of input atmospheric and geometric variables according to the selected option. Three LUT node distribution methods are implemented in ALG. The first method corresponds to a systematic (gridded) combination of all input variables and their values. Assuming $D$ selected input variables, each of them with $p_i$ values ($i$=1 to $D$), the output LUT will contain $N = \prod_{i=1}^{D} p_i$ nodes. The second method correspond to a pseudo-random distribution of nodes homogeneously covering the $D$-dimensional input space with a user-defined $N$ scattered nodes. The final method is based on an automatic node distribution algorithm, GALGA, that minimizes the error in the linear interpolation of simulated TOA radiance below a user-defined error threshold value. This gradient-based node distribution has shown to reduce interpolation errors by at least 10% and LUT size (and thus computation time) by at least 25% (Vicent et al., 2018). ALG includes a multi-dimensional interpolation function that works both with gridded and scattered data. The implemented LUT interpolation methods involve: (1) nearest neighbour, (2) piece-wise linear (Abramowitz and Stegun, 1964), (3) piece-wise cubic splines (Bartels and Barsky, 1998), (4) inverse distance weighting (Shepard, 1968), and (5) $D$-dimensional triangulation (Delaunay, 1934; Barber et al., 1996).

In step 2a, the LUT generation process continues by converting the determined combinations of atmospheric/geometric variables and user-input spectral configuration into a set of RTM input files required to build the atmospheric LUT. In this step, ALG detects if the user has selected any default or user-defined OPAC aerosol model. If so, ALG automatically runs OPAC and saves the output aerosol optical properties for a later use (see step 2b). Following the approach proposed in (Huang et al., 2016), the values of these aerosol properties, spectral configuration and additional atmospheric input variables (i.e., the LUT nodes) are written in $P$ subsets of RTM input files. In step 3 (*Run atmospheric RTM*), parallel instances of the selected RTM are then run in batch mode based on these input files.

In step 4, and once all the RTM simulations are correctly executed, ALG will finalise the LUT generation process by reading, processing and storing the RTM output data files in the final LUT file. One of the key aspects of ALG is that it harmonizes the variety of RTM spectral outputs into a common and consistent definition of the stored LUT data. For this, ALG uses the so-called *atmospheric transfer functions*, typically used in remote sensing applications. These atmospheric transfer functions

permit uncoupling the radiative transfer effects of the between the surface and atmosphere and thus are particularly useful in atmospheric correction and forward modeling (Vermote et al., 1997; Matthew et al., 2000; Guanter et al., 2009; Verhoef and Bach, 2012). In the case of a Lambertian and homogeneous surface with reflectance $\boldsymbol{\rho}$, a TOA radiance spectrum ($\boldsymbol{L}_{toa}$) can be calculated through Eq. (1):

$$\boldsymbol{L}_{toa} = \boldsymbol{L}_0 + \frac{(\boldsymbol{E}_{dir}\mu_{il} + \boldsymbol{E}_{dif})(\boldsymbol{T}_{dir} + \boldsymbol{T}_{dif})\boldsymbol{\rho}}{\pi(1 - \boldsymbol{S}\boldsymbol{\rho})} \tag{1}$$

where $\mu_{il}$ is the cosine of the SZA. The LUTs generated by ALG contain the atmospheric transfer functions used in Eq. (1) and which are described below:

- The spectrum of intrinsically reflected radiance by the Earth's atmosphere ($\boldsymbol{L}_0$ in $\mathrm{mW} \cdot \mathrm{m}^{-2}\mathrm{sr}^{-1}\mathrm{nm}^{-1}$), also called atmospheric path radiance.

- The downwelling solar irradiance spectrum at surface level, splitted by its direct ($\boldsymbol{E}_{dir}$) and diffuse ($\boldsymbol{E}_{dif}$) fluxes, both in $\mathrm{mW} \cdot \mathrm{m}^{-2}\mathrm{nm}^{-1}$.

- The atmospheric reflectance spectrum for the photons backscattered to the surface ($\boldsymbol{S}$), also known as spherical albedo.

- The upwelling direct and diffuse target-to-sensor transmittance spectra ($\boldsymbol{T}_{dir}$ and $\boldsymbol{T}_{dif}$).

In addition to these atmospheric transfer functions, the generated LUT file also includes:

- The extraterrestrial solar irradiance spectrum at 1AU Earth-to-Sun distance, $\boldsymbol{I}_0$ in $\mathrm{mW} \cdot \mathrm{m}^{-2}\mathrm{nm}^{-1}$.

- The wavelength vector at which these spectral magnitudes are calculated.

- The name and values of the input atmospheric and geometric variables for each LUT node.

- The values of the remaining (constant) parameters.

An important part of the complexity of ALG lies in being able to harmonize the different radiative transfer codes, with different types of outputs, to fill the exact same LUT. For MODTRAN simulations, these spectrally-dependent atmospheric transfer functions are automatically calculated by applying the interrogation technique presented in Guanter et al. (2009) and Verhoef and Bach (2012). In the case of libRadtran simulations, four runs are needed to compute these transfer functions (Debaecker et al., 2016). Similarly, 6SV directly provides the atmospheric transfer functions, however, with a slightly different definition due to the uncoupling of scattering and gas transmittance. The following transfer functions are used for 6SV: path radiance, at-surface total solar irradiance due to scattering ($\boldsymbol{E}_{tot}$ in $\mathrm{mW} \cdot \mathrm{m}^{-2}\mathrm{nm}^{-1}$), total gas transmittance ($\boldsymbol{T}_{gas}$), total upwelling transmittance due to scattering ($\boldsymbol{T}_{tot}$) and spherical albedo ($\boldsymbol{S}$). In this case, $\boldsymbol{L}_{toa}$ is calculated through Eq. (2):

$$\boldsymbol{L}_{toa} = \boldsymbol{L}_0 + \frac{\boldsymbol{T}_{gas}\boldsymbol{E}_{tot}\boldsymbol{T}_{tot}\boldsymbol{\rho}}{\pi(1 - \boldsymbol{S}\boldsymbol{\rho})} \tag{2}$$

**Table 1.** Key input atmospheric variables used in MODTRAN5, libRadtran and 6SV to perform the GSA. Atmospheric profile was set to US Standard 1962 (Anderson et al., 1986).

| Variable name | Min-Max |
| --- | --- |
| Elevation (h): | 0 - 3 km |
| Aerosol optical thickness (AOT): | 0.05 - 1 |
| Ångström exponent ($\alpha$): | 0.1 - 1.5 |
| Aymmetry parameter (G): | 0.6 - 1 |
| Single Scattering Albedo (SSA): | 0.75 - 1 |
| Water vapor (CWV): | 1 - 4 $\mathrm{g \cdot cm^2}$ |
| Ozone (O3): | 0.25 - 0.45 $\mathrm{atm-cm}$ |

## 4  Model intercomparison

As a first step for the RTM intercomparison study, we carried a global sensitivity analysis (GSA) of atmospheric RTM simulations. GSA allows to identify the key input variables driving the spectral output and variables of lesser influence. By identifying variables of lesser influence, models and generated LUTs can be greatly simplified, which facilitates applications such as inversion of biophysical parameters and atmospheric correction. In short, sensitivity analysis algorithms determine the effect of changing the value of one or more input variables, and observing the effect that this has on the RTM output. GSA, where the role of all input variables and their interactions are analyzed, have been successfully applied in vegetation and atmospheric RTMs (Verrelst et al., 2016; Vicent et al., 2017).

Here, we used ALG to generate a set simulations in order to analyse the relative impact of key atmospheric variables into TOA radiance. Three LUTs of MODTRAN5, libRadtran and 6SV simulations were generated. They consist of 2000 samples distributed with a Latin Hypercube sampling and covering the entire 400-2500 nm spectral range at 15 $\mathrm{cm^{-1}}$ (0.24-9 nm) for MODTRAN and libRadtran and 2.5 nm for 6SV. These LUTs vary the atmospheric conditions as summarized in Table 1, with geometry fixed to SZA=30°, VZA=0° and a relative azimuth angle (RAA) of 0°.

The generation of RT model input files is straighforward with ALG: the range of input variables given in Table 1 are introduced by the user through ALG's interface. ALG processes this input configuration and prepares the input files according to the user manual of each RT model for their specific format. For MODTRAN5 and libRadtran, all the input variables are actual parameters of these models as specified in the respective user manuals. However, for 6SV, the introduction of aerosol optical properties $\alpha$, G and SSA is achieved through the preparation of a specific 6SV `.mie` file. The reader should notice some of the main differences between the compared models as highlighted in Table 2 in order to support the later discussion about the observed differences. For all the 2000 combinations, the atmospheric transfer functions generated by ALG were coupled with a typical vegetation spectrum simulated with PROSAIL model (Jacquemoud et al., 2009) based on Eq. (2) using ARTMO's TOC2TOA toolbox (Verrelst et al., 2019).

**Table 2.** Key commonalities and differences between MODTRAN5, libRadtran and 6SV simulations.

| Feature | MODTRAN5 | libRadtran | 6SV |
|---|---|---|---|
| RT solver: | DISORT | DISORT | Successive Orders of Scattering |
| Absorption modelization (resolution): | Correlated-k ($15\ \mathrm{cm}^{-1}$) | Reptran ($15\ \mathrm{cm}^{-1}$) | Band model (2.5 nm) |
| Coupled absorption-scattering (Yes/No): | Yes | Yes | No |
| Aerosol optical prop. (input config.): | Input parameters (i.e., $\alpha$, G, SSA) | | Precalculated through `.mie` file |
| Aerosol optical prop. (vertical distr.): | Only in boundary layer | | Optical properties common for entire column |

Before analysing the GSA results, we illustrate in Figure 4 the path radiance, spherical albedo and total solar irradiance
calculated by the three selected atmospheric RTMs. In this figure only 16 spectra are shown, corresponding to all the min/max
values of the 4 aerosol parameters given in Table 1 in order to illustrate the full variance in the database. The sub-axis zoom in
the spectral window between 750-860 nm where the absorption features of the $O_2$-A and $H_2O$ are visible.

This Figure 4 illustrates the consistent MODTRAN, libRadtran and 6SV simulations achieved with the use of ALG. Overall,
it is observed how the three spectral magnitudes are overlapping in the entire 400-2500 nm spectral range. We can also observe
that approximately six spectra out of the 16 plotted spectra are mostly visible, which indicates that only two variables dominate
the entire variance of the signal as it will later be discussed through the GSA analysis. Despite the agreement of the various
RTMs, some discrepancies appear in the figures. Firstly, regardless of the spectral resolution, we find that 6SV has a better
agreement with libRadtran than with MODTRAN5. The disagreement with MODTRAN5 is particularly higher at higher path
reflectances and lower transmittances, which might indicate that MODTRAN tends to increase the effect of scattering through
the phase function with respect to libRadtran and 6SV. Secondly, it is clearly observed how the spherical albedo in 6SV
simulations is free of gas absorptions. Indeed, this is a result of the decoupling of gas absorption from scattering by molecules
and aerosols in 6SV. Lastly, there are minor differences in the spectral features of the gas absorptions, which can be due to the
absorption modelization (correlated-k in MODTRAN and Reptran in libRadtran) as well as differences in spectral resolution
(2.5 nm in 6SV and $15\ \mathrm{cm}^{-1}$ in MODTRAN and libRadtran.

The comparative analysis is followed in Figure 5 through the total sensitivity index (SI), which shows the relative importance
of each input variable at TOA radiance for typical vegetation spectrum.

In general, all three RTMs show similar GSA results, indicating that they simulate similarly the processes of absorption
and scattering. In these models, the driving variables are those related with the aerosol particles (AOT, $\alpha$, G and SSA), which
cause the scattering and thus path radiance and diffuse transmittance along the entire spectral range. The Ångström exponent
increases its relative importance as wavelength increases from 550 nm, which is the anchor wavelength at which the AOT
is defined. The surface altitude has its major influence ($\sim 80\%$) at the bottom of the $O_2$-A absorption ($\sim$760 nm) since the
aborption is mostly driven by the surface pressure. As expected, the importance of CWV is localized at the specific wavelengths
of $H_2O$ absorptions. All models also show a sudden decrease of the relative importance of the scattering processes (through the

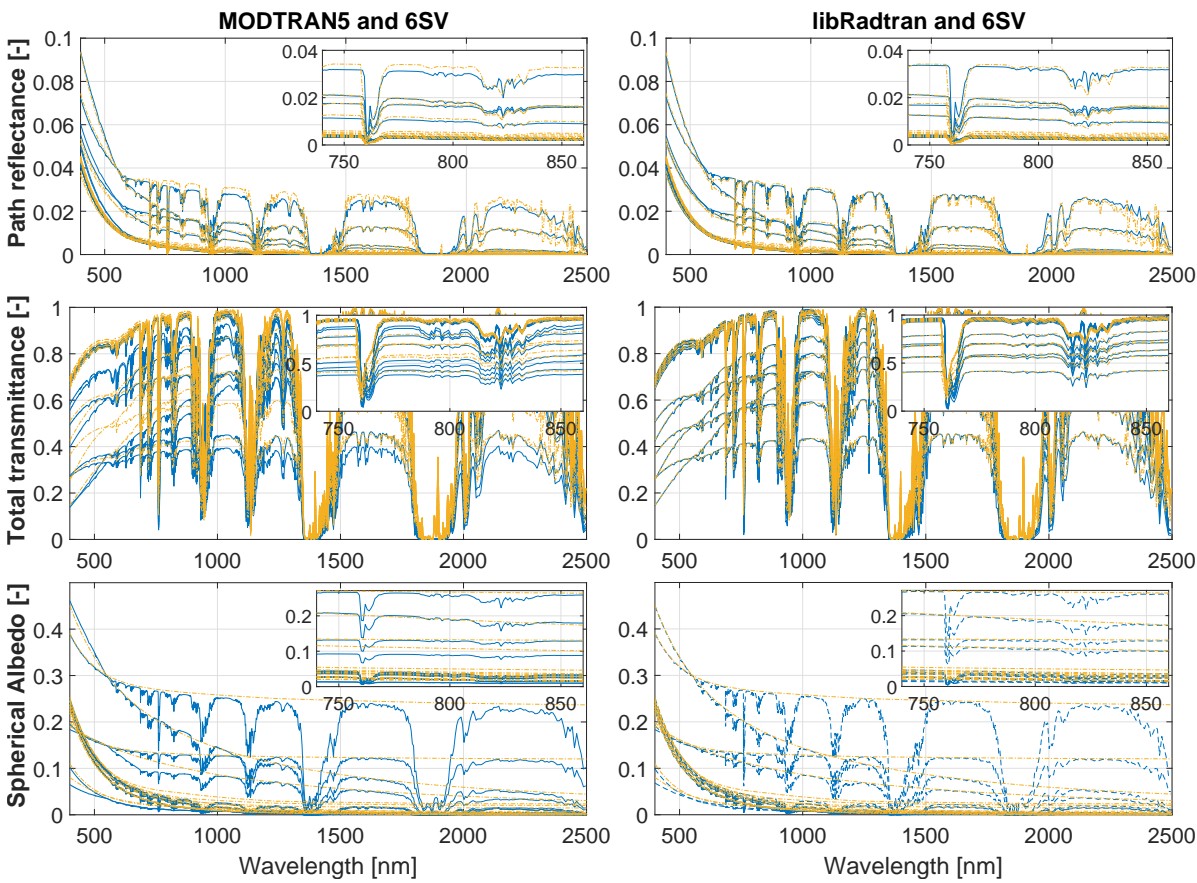

**Figure 4.** Path reflectance (top), total transmittance (mid) and spherical albedo (bottom) spectra comparison between MODTRAN5 and libRadtran (blue, left and rigth columns respectively) and 6SV (yellow).

variables G and AOT) after $\sim$720 nm. Indeed, according to Eq. (1), the high reflectance values of vegetation in the near infrared spectral region reduce the influence of the atmospheric path radiance (most affected by the scattering processes) with respect to the surface-reflected radiance. Despite of these similarities, the GSA figures also show some discrepancies, particularly on the lower importance of the aerosol absorption (through the SSA variable) in MODTRAN5 for wavelengths higher than $\sim$720 nm. The MODTRAN5 model also shows some sensitivity ($5-10\%$) to the asymmetry asymmetry parameter (G) in the 720-1300 nm spectral range while it is nearly 0% in libRadtran and 6SV, in agreement with our observations in Figure 4. Important differences also appear on the relative sensitivity of surface elevation and CWV within the $H_2O$ bands. In fact, both variables compete to influence the strength of the $H_2O$ absorption, the CWV through its influence on the amount of $H_2O$ in the atmospheric column and surface elevation directly on the definition of the optical path of photons. In this case, 6SV shows higher dependency on the surface elevation than MODTRAN and libRadtran due to uncoupled scattering and absorption effects in 6SV. In 6SV, the $H_2O$ absorption only affects to the direct Sun-target-sensor transmittance component, which is dependent on both the CWV and optical path (and thus surface elevation). In MODTRAN and libRadtran, the multiple

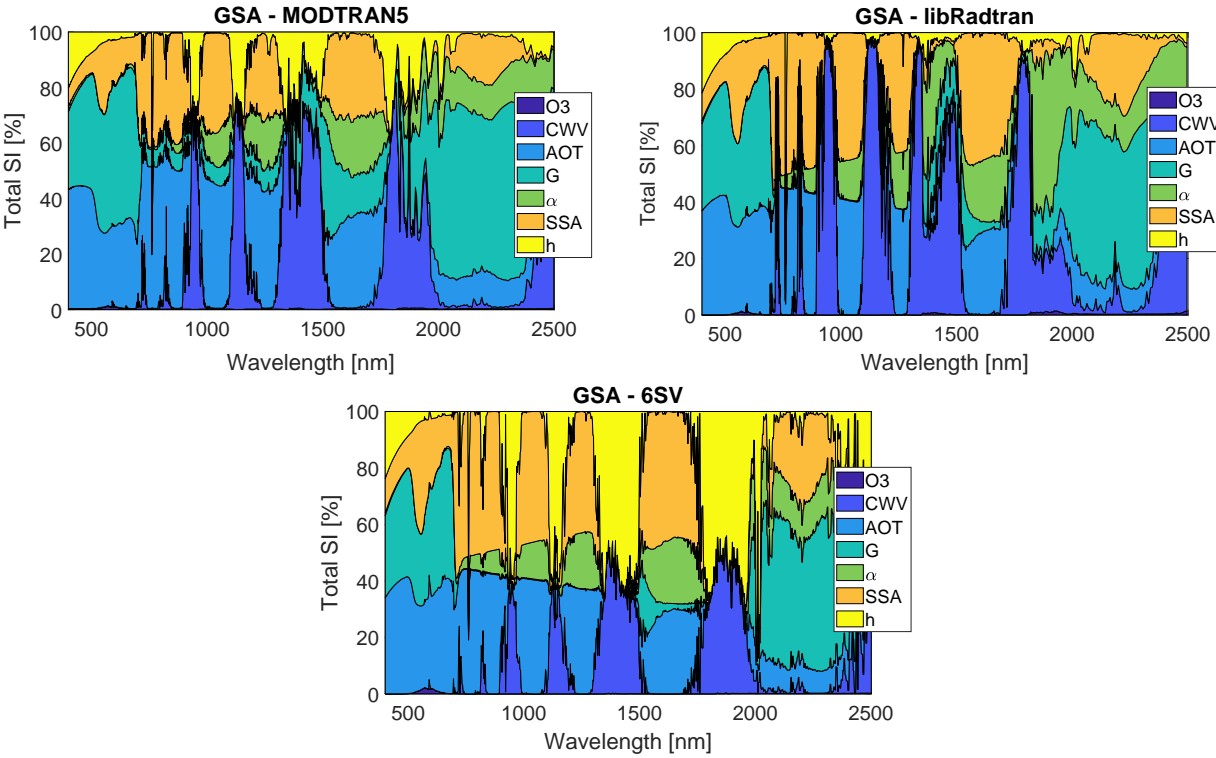

**Figure 5.** MODTRAN5, libRadtran and 6SV GSA results of main atmospheric properties at TOA radiance.

scattering increases the optical path of photons and thus the absorption by $H_2O$, which makes the model more sensitive to the CWV than surface elevation. However, MODTRAN and libRadtran still show differences in the relative sensitivity to CWV versus surface elevation, which indicates differences in the implementation of the coupled absorption-scattering processes at these strong absorption features or the definition of the scattering properties. In fact, the aerosol optical properties (i.e., $\alpha$, G and SSA) in MODTRAN are defined for the boundary layer aerosols while in libRadtran and 6SV they are common for the entire column.

To further prove the usefulness of ALG to perform RTM intercomparison, we secondly repeated the study in (Kotchenova et al., 2008). Here, we compared 6SV simulations against MODTRAN's DISORT (Stamnes et al., 1988) (8 streams) and Isaac's 2-streams (Isaacs et al., 1987) RT solvers and libRadtran (DISORT solver with 8 streams). The simulations were performed with a US Standard 1976 atmospheric profile, the OPAC's continental average aerosol model optical properties, two values of AOT (0.2 and 0.8) and the same range of illumination/observation conditions described in (Kotchenova et al., 2008). The

simulated LUTs were used to calculate the intrinsic atmospheric reflectance. The atmospheric reflectance from MODTRAN and libRadtran ($\rho'$) was compared with the simulated by 6SV ($\rho'_{6sv}$) according to the following cost function:

$$\delta(\tau, VZA, \lambda) = \frac{100}{N} \sum_{SZA} \sum_{RAA} \frac{|\rho'_{6sv} - \rho'|}{\rho'_{6sv}} \tag{3}$$

320     where, for sake of simplicity, we have omitted the dependency of the reflectance with the AOT ($\tau$), VZA, SZA, RAA and wavelength ($\lambda$). Figure 6 shows the results of the average relative differences for two wavelenghts ($\lambda$=412 nm and 670 nm).

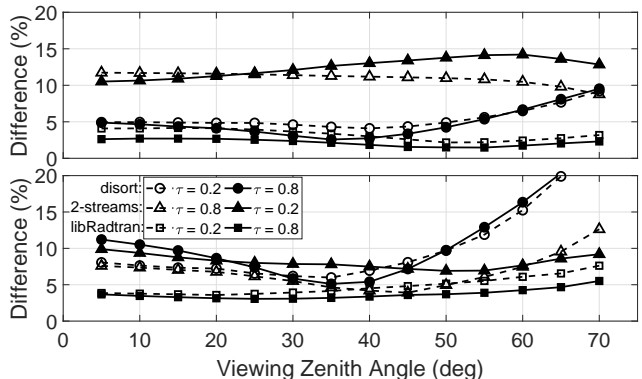

**Figure 6.** Average relative differences between 6SV and MODTRAN DISORT, Isaac's 2-streams and libRadtran DISORT for two wavelengths, 412 nm (top) and 670 nm (bottom), as function of VZA and AOT.

The results are compatible with those presented in (Kotchenova et al., 2008), showing differences (at 412 nm) of 5-10% with respect to 6SV mostly due to the simulation of polarization in 6SV and the calculation of multiple-scattering by the Henyey–Greenstein aerosol phase function. These effects are also seen when using the Isaac's 2-streams radiative solver in

325    MODTRAN, now with errors up to 15%. The discrepancies with respect to libRadtran are rather constant, with errors around 3-4%, probably since libRadtran introduces the phase function calculatd by OPAC for the simulation of scattering effects.

## 5    Other applications

As described in the section 3, ALG facilitates the usage of atmospheric RTMs and the generation of large LUTs of atmospheric transfer functions. Users can integrate these LUTs into a wide range of applications.

330    One of these applications is in End-to-end mission performance simulators (E2ES). E2ES are software tools that reproduce all aspects of satellite missions including the platform orbit/attitude, synthetic scene generation, sensor behavior, ground image processing and product evaluation (Kerekes et al., 1999; Segl et al., 2012). These tools are used by remote sensing scientists and engineers to support trade-off studies, to prepare of system calibration tests and to optimize data processing algorithms. As part of the European Space Agency's FLEX E2ES (Vicent et al., 2016), pre-computed MODTRAN-based LUTs generated

335    with ALG are used to simulate the radiance signal as would be observed by FLEX mission instruments (Tenjo et al., 2017).

Another typical application of atmospheric LUTs is on the retrieval of aerosol physical and optical properties (Dubovik et al., 2002; Huang et al., 2015). From a satellite data processing perspective, aerosols are one of the main atmospheric components that must be accounted for when performing atmospheric correction (Thompson et al., 2018). In this frame, we studied the impact of aerosol type variability in the atmospheric correction within the $O_2$ absorption regions (Vicent et al., 2017). The goal was to determine whether the use of parametric approximations in aerosol properties can be used to peform the atmospheric correction in the $O_2$ absorptions. ALG was used to simulate several datasets with varying aerosol types, optical properties and vertical distribution.

The applicability of ALG is not limited to spaceborne instruments, but they is also suitable for the analysis of airborne and proximal sensing (e.g., flux towers, unmanned aerial vehicles). In our recent publication (Sabater et al., 2018), we studied the impact of path length in proximal sensing measurements of downwelling irradiance and at-sensor radiance, and their impact on sun-induced fluorescence retrieval. The study focused on remote sensing instruments placed at 2-50 m height over the surface. ALG was used to facilitate the running of MODTRAN simulation, which varied the instrument height and the SZA for standard atmospheric conditions.

Altogether, these few examples demonstrate the versatility of ALG to address multiple remote sensing applications based on the use of atmospheric RTMs.

## 6 Conclusions and future work

In this paper the main design concept and features of ALG have been described along with an intercomparison study for the atmospheric RTMs 6SV, MODTRAN and libRadtran. The a priori tedious tasks of (1) writting consistent input files, (2) running the RTMs in an efficient manner, (3) compiling and harmonizing the various model output files into ready-to-use LUT files, and (4) perform a model sensitivity analysis was largely simplified using the developed ALG tool and its compatibility with the ARTMO software framework (Verrelst et al., 2012). The sensitivity analysis results indicate that, overall, the various atmospheric RTMs simulate similarly the absorption and scattering processes for the selected atmospheric variables. However, there are still important differences in the sensitivity analysis that must be analysed in more detail.

Other practical applications, such as scene generation, atmospheric data analysis and atmospheric correction (Thompson et al., 2018), can also benefit from the use of ALG. A few application examples were presented, demonstrating the software capabilities to generate consistent LUTs for several atmospheric RTMs, with a wide range of input atmospheric variables, nodes distribution and spectral configurations. ALG is an ongoing work and regularly updated with new added functionalities and tools. The following upgrades are in the pipeline: (1) including the polarization data calculated by the 6SV code and the polRadtran and Mytic solvers in libRadtran, (2) implement functions to develop emulators of atmospheric transfer functions, (3) generation of LUTs of TOA radiance spectra for non-Lambertian and non-homogeneous surfaces, (4) implementation of additional RTMs such as RRTOV and SOS, and (5) compatibility with Linux and MacOS systems. Summarizing, ALG can become an useful tool to facilitate research on atmospheric radiative transfer, as well as opening the use of atmospheric RTMs

to wider research communities and applications such as for climate studies, atmospheric physics and chemistry and remote sensing data processing.

*Code availability.* The exact version of the ALG (v2.0) used to produce the results used in this paper is archived on Zenodo (https://doi.org/10.5281/zenodo.3555575), as are input data and scripts to run the model and produce the plots for all the simulations presented in this paper. The current version of ALG is freely available from the project website (http://ipl.uv.es/artmo) under the GNU General Public License v3 (see http://www.gnu.org/licenses/). The software package has been developed in Matlab® R2018a and it is compatible with Windows operating systems. The tool is also provided as a stand-alone compiled executable file so that users not having a licence of Matlab
can still run the software. Accordingly, users must first install the corresponding Matlab Runtime (MCR version 9.5, 64-bits). In addition, the help system of ALG includes a set of guidelines to install and compile the compatible atmospheric RTMs. The user should notice that ALG does not re-distribute the source code nor the compiled version of the underlying third-party atmospheric RTMs due to license rights.

*Author contributions.* JVi and NS designed the ALG tool. JVi, JRC and JMM developed the ALG tool and performed the simulations. JVi and JVe designed the simulation experiments and carried them out. JVi prepared the manuscript with contributions from all co-authors.

*Competing interests.* The authors declare that they have no conflict of interest.

*Acknowledgements.* This work was partially carried out in the frame of the Spanish R&D plan project *Advanced L3-L4 products for the FLEX-S3 mission* no. RTI2018-098651-B-C51 and ESA's project *FLEX L1B to L2 Algorithms Development study* with ESA-ESTEC contract no. AO/1-8897/17/NL/MP. Jochem Verrelst was supported by the European Research Council (ERC) under the ERC-2017-STG SENTIFLEX project (grant agreement 755617).

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
