# Peer review of "Comparative analysis of atmospheric radiative transfer models using the Atmospheric Look-up table Generator (ALG) toolbox (version 2.0)"

_Geoscientific Model Development, 2019_

## Short Comment (SC1) · 15 Oct 2019

Hello Jorge,

Could you please indicate which type of parametrization approach was used in the simulation performed with libRadtran. Was it LOWTRAN, REPTRAN (fine, medium, coarse)? And at which spectral resolution?

Best regards, Jérôme LOUIS
* * *

---

## Author Comment (AC1) · 15 Oct 2019

Dear Jérôme, On behalf of all the co-authors, please let me thank you for posting your comment, which I hope is duly addressed here. In our comparative study, libRadtran was configured with the REPTRAN coarse (at 15 cm-1 spectral sampling) parameterization for molecular absorption, which is the default option used by libRadtran. LibRadtran accepts other spectral samplings for REPTRAN: medium (5 cm-1) and fine (1 cm-1). Other parameterizations are also implemented in libRadtran, such as LOWTRAN, the integrated bands of Kato and Fu, and the possibility of directly using molecular cross-sections and optical deppth for higher spectral resolution. From libRadtran's

user manual, the REPTRAN parameterization is recommended for general user-case applications: "Though we recommend REPTRAN for spectral calculations, the molecular absorption pa-rameterization from LOWTRAN/SBDART by Ricchiazzi et al. (1998) is available mainly for compatibility reasons" In ALG, the current options are REPTRAN (3 resolution options) and LOWTRAN. The implementation of molecular cross-section is work in progress. Kind regards, Jorge

---

## Referee Comment (RC1) · Anonymous Referee #1 · 21 Oct 2019

OVERVIEW

The authors describe a model that can have utility for inter-comparing radiative transfer models. There is clearly a scientific community that could use such a model. The authors present a sound and systematic approach to addressing this need. The material is topical to the journal it has been submitted to.

With that said, I feel there is a pressing need for a substantial reorganization of Section 3. The authors present two figures to illustrate the programmatic flow of their ALG model. Those flow diagrams define a natural process for describing the functionality of

the model but the text didn't clearly follow the diagrams. As a reader new to this model, I found myself constantly jumping between the prose and the figures and working to determine how the text connects to the process. As an example, consider Figure 2 ("LUT Configuration"). I presume this is the same as the "LUT config." box referenced in the GUI box of figure 1, but it's never explicitly stated. Then, within Figure 2 itself, there are numbered boxes, but those numbers are never explicitly made use of in the prose: either omit the numbers because they don't add value to your discussion, or mention them in the text because they do.

SECTION 3 PROPOSAL

I'd like to propose the following sequencing:

1. consider turning Figure 1 into a high-level architecture diagram with just a few boxes (the three boxes currently in GUI, the mention of an output config file, and a single box representing "Internal Functions"). I count five potential boxes: mention each of the functions of these five boxes in a logical sequence. For two of those boxes (LUT config and Internal Functions), you can be very brief and conclude with "We will discuss LUT Configuration in greater detail in section 3.x"

2. Then, take your existing Figure 2 ("LUT Configuration") and give a brief, sequential discussion of the purpose of each of the five boxes. The text is mostly already in the manuscript, but should be organized around the figure itself. The figure has five boxes: I should see five paragraphs - or, more generally, five logical groupings of information. Finally, the bottom half of your Figure 1 ("ALG - Internal Functions") can be turned into it's own Figure 3. It should follow Figure 2 in the sense that you have to configure before you can run. Once again, five boxes means five paragraphs. Some of this text will be new and some can be derived from the existing section 3.

3. You will end up with some text that doesn't fit neatly into the flow but you likely want to keep (because you're making important points about ALG). This content can appear as a lead-in to the section (before Figure 1), as concluding remarks about the benefit of this approach (after describing Figure 3), or as part of your concluding remarks

The authors are certainly free to take a different approach if they feel there is a better way to get at the heart of the problem. Regardless, the authors need to help the reader, presumed to be totally new to ALG, with a logical walk-through of what this model does and why.

ADDITIONAL COMMENTS

1. Lines 161-168 + Lines 198-200: This discussion belongs somewhere else. You're telling the user that once you've built the LUT on a discrete grid, you can interpolate to any continuous value within that grid without losing much accuracy. That information should appear in the paper, but Section 3 really focuses how the model gets constructed (not used). I'd propose deferring this paragraph until you've completed your description of how the LUT was created in the first place. The second set of lines discusses details of how the interpolation occurs. If you aggregate these two blocks, you end up with a useful description of how ALG gets used in real-world situations.

2. Lines 207-218 + Lines 230-232: This is an important paragraph that tells the user exactly what you store in your LUT. Give motivation as to why these particular quantities. Also, you list several quantities in the LUT as bullet points, then switch gears to discuss a technical issue that differs between codes, and then back to completing LUT quantities, this time as numbered items. Why not complete the bulleted list first and then follow up with the difficult LUT entry. Finally, this is the

spot where the authors have the opportunity to tell the reader the hard work that has clearly been done to make different radiative transfer codes, with different types of outputs, fill the exact same LUT.

3. Lines 219-230: The text is fine, but please lead in with a motivating sentence. Something like "Most values stored in the ALG LUT can be obtained directly from standard RTM outputs. The exception is TOA radiance: obtaining this value differs depending on the RTM being used."

4. Figure 4: Please define "SI" somewhere (in the figure or in the text)

COMMENT TO EDITORS

Some symbols did not print for me (e.g., a lot, but not all, capital "A" values were missing. This could be a quirk of my system and there might be nothing wrong with the PDF. Just in case, I ask the editors to perform a typographic check prior to publication.

---

## Referee Comment (RC2) · Anonymous Referee #2 · 18 Nov 2019

The paper describes a software tool to generate lookup tables for atmospheric correction. Three different RT models (MODTRAN, libRadtran and 6SV) are included and the tool provides an interface which can setup consistent model input files for the different RT models. Thus the tool can be used to compare the different models, which is an application shown in the paper.

The paper is a technical description of the software tool ALG. Unfortunately. the results of the model intercomparison shown here are not very meaningful, because the authors do not explain the reasons for descrepencies, they only speculate and do not provide details of the RT codes. Moreover, I think that the compared quantity (global

sensitivity analysis) is not well suited for a model intercomparison, because it does not give much insight in reasons for descrepencies. All included RT models (MODTRAN, 6SV and libRadtran) have been validated in numerous model intercomparison studies and given the same input they produce exactly the same results. Providing exactly the same input is a difficult task, and obviously, also the presented toolbox can not generate exactly the same input. There are differences in parameterisations of absorption, optical properties etc., so I think that it is important, that one works directly with the RT model instead of using a wrapper which hides the specific features of the individual models.

I am a little skeptical whether the tool is really useful for the scientific community. Of course it makes the generation of lookup-tables for atmospheric correction easier. But the danger is that the users do not understand the physics behind radiative transfer, which they learn better when they work directly with the RT models.

Given these concerns, I can not recommend publication in GMD. The document is of course needed as a technical documentation of the software but it's scientific content is from my point of view not sufficient for a scientific journal such as GMD.

---

## Short Comment (SC2) · 26 Nov 2019

This is an executive editor comment highlighting the ways in which this manuscript is not currently compliant with GMD policy on code and data availability. Currently the manuscript fails to comply with very significant parts of the GMD requirements for source code publication. If these issues cannot be remedied then the manuscript will need to be rejected.

GMD requires the public and persistent archiving of all of the source code on which a manuscript depends. This means that only releasing binary code, as appears to be the case for ALG, is incompatible with publication in GMD. The full description of

this policy and its rationale is presented in the recent editorial (https://doi.org/10.5194/gmd-12-2215-2019), though that is merely the most recent restatement of the principle, which has always been the GMD approach. The only exceptions to this rule occur where it is impossible to release the source code for reasons beyond the authors' control. This is typically only the case for large models owned by national institutions, where redistribution policies are controlled at a level far removed from the scientists who write the code and papers. In this case, both of the researchers named as authors of the software on the software package's website are authors of the manuscript, and the licence conditions of the software appear to be the result of the participants' own choices. In these circumstances, publicly and persistent archiving the source code employed is a necessary precondition for publishing in GMD.

Full details of these requirements are available on the GMD website in the code and data availability policy: https://www.geoscientific-model-development.net/about/code_and_data_policy.html.

———————————————————

---

## Author Comment (AC4) · 29 Nov 2019

Dear Executive Editor Dr. David Ham,

First of all, we would like to thank you for informing us about the GMD policy on code and data availability. We realized that that indeed our paper was not correctly fulfilling the GMD policy as also indicated in the recent editorial note (https://doi.org/10.5194/gmd-12-2215-2019) and the full details on the requirements for data and code availability (https://www.geoscientific-model-development.net/about/code_and_data_policy.html).

[Figure]

We have carefully reviewed the information in these websites as well as the recommendations you kindly send us by personal e-mail. Accordingly, we have made available the source code of ALG v2.0 (used in this paper) as well as the generated data and scripts using the storage provided by Zenodo under the following link: **https://doi.org/10.5281/zenodo.3555575**

We have also updated the manuscript on the *Code availability* section with the following text:

*The exact version of the ALG (v2.0) used to produce the results used in this paper is archived on Zenodo (https://doi.org/10.5281/zenodo.3555575), as are input data and scripts to run the model and produce the plots for all the simulations presented in this paper. The current version of ALG is freely available from the project website (http://ipl.uv.es/artmo) under the GNU General Public License v3 (see http://www.gnu.org/licenses/). The software package has been developed in Matlab® R2018a and it is compatible with Windows operating systems. The tool is also provided as a stand-alone compiled executable file so that users not having a licence of Matlab can still run the software. Accordingly, users must first install the corresponding Matlab Runtime (MCR version 9.5, 64-bits). In addition, the help system of ALG includes a set of guidelines to install and compile the compatible atmospheric RTMs.*

The source code is therefore freely available for the community through the Zenodo repository as well as in our own project website (www.artmotoolbox.com).

We hope that these modifications are now inline with the GMD policy on source code and data availability and thus that our manuscript is compliant for its publication in GMD. Please don't hesitate to contact us if you have any further comments or remarks.

On behalf of the co-authors, I would like to thank you once more for your support.

Kind regards,

The authors

---

## Author Comment (AC5) · 2 Feb 2020

Dear anonymous reviewer, I would like to further insists on our view of the usefulness of our ALG tool. I have recently assisted to the workshop Trattoria 2020 (http://www.meteo.fr/cic/meetings/2020/trattoria/), organized by CNES and Meteo France. This workhop is oriented to atmospheric radiative transfer for atmospheric physics, atmospheric correction and climate modelling. In this workshop I had the chance to show ALG to model developers (e.g. SOS-abs, Artdeco, 4AOP,...) and I generally got a positive feedback on the interest of ALG by the community. The interest came not only for the possibility of facilitating the generation of look-up tables but

also the possibilities offered for facilitating model intercomparison or preparing data for practical applications (e.g. for atmospheric correction and satellite data processing algorithms). ALG does not replace the RT models but support and facilitate their use. We therefore consider that our toolbox can be useful to understand the RT models and physics.
* * *

---

## Author Response (AR1)

**Authors' response to Reviewers**

**Response from the Authors to** the comments from the **Anonymous Referee #1**:

*Dear anonymous referee*

*First, thank you for your interest in our publication and in the ALG software tool. We also believe that a broader scientific community might be interested in using the provided tool. We kindly appreciate your comments and advices, which have served us to improve the quality of the manuscript.*

*Regarding your comments, we have been taken them carefully into consideration. Specifically, we have reviewed our paper, trying to improve the description of the functionality of the model. Please find in the table below the main changes of our paper, which are also highlighted in red in the updated manuscript to facilitate the revision process.*

*We trust that the comments raised through the revision process have been duly addressed to your satisfaction.*

*Kind regards,*

*The Authors*

**Reviewer #1 comment #1 (1.1): I feel there is a pressing need for a substantial reorganization of Section 3**

We appreciate your comment and the remark on the reorganization of Section 3. After reading your comments, we have made a substantial change on the structure of Section 3. We think that, with these changes, the concepts behind Section 3 have been clarified. Please observe the changes (in red) in our manuscript in Section 3. Particularly, see our replies to your specific comments in the following comments.

**1.2: The authors present two figures to illustrate the programmatic flow of their ALG model. Those models flow diagrams define a natural process for describing the functionality of the model byt the text didn't clearly follow the diagrams [...] As an example, consider Figure 2 ("LUT Configuration"). I presume this is the same as the "LUT config." box referenced in the GUI box of figure 1, but it's never explicitly stated.**

We acknowledge that the old Figure 1 might be confusing on the logic and purpose of the Section 3.2 (i.e. to describe ALG's graphical user interface). We have now modified Figure 1 (page 6) to focus only on the graphical user interface. We also modified the caption in Figure 2 caption (also in page 6) to explicitly state this link between "LUT configuration" boxes.

**1.3: Within Figure 2 itself, there are numbered boxes, but those numbers are never explicitly made use of in the prose: either omit the numbers because they don't add value to your discussion, or mention them in the text because they do.**

We understand the comment from the Referee. However, the numbers are already used explicitly in the prose and each box has a dedicated paragraph. See "Firstly" (line 154 using previous version numbering), "Secondly" (line 162), "Thirdly" (line 169), "Fourthly" (line 172)

and "Finally" (for the 5th step; line 175). In order to improve readability and link with the numbers in Figure 2, we have now changed those ordinal numbers (i.e., firstly, secondly…) by expressions such as "In step 1 (Generic configuration), …" (see lines 155, 162, 169, 173 and 177 in the updated manuscript).

We believe that, with this change, it is more explicit and clear the link between the steps in Figure 2 and the paragraphs below.

***1.3: (Section 3- Proposal) Consider turning Figure 1 into a high-level architecture diagram with just a few boxes (the three boxes currently in GUI, the mention of an output config file, and a single box representing "Internal Functions"). I count five potential boxes: mention each of the functions of these five boxes in a logical sequence. For two of those boxes (LUT config and Internal Functions), you can be very brief and conclude with "We will discuss LUT Configuration in greater detail in section 3.x"***

We appreciate the recommendation given by the Referee. We realize that the key problem in understanding and following the logic of ALG is on the position of Figure 1 within Section 3.2 (*ALG graphical interface*). We have decided to update Figure 1 (see page 6 of the updated manuscript), only including the three blocks of the graphical user interface: Software configuration, LUT configuration and Help system. These three blocks are then described in Section 3.2 in lines 150-153 (Software configuration), lines 154-188 (LUT configuration) and lines 190-193 (Help system).

***1.4: (Section 3 – Proposal) Then, take your existing Figure 2 ("LUT Configuration") and give a brief, sequential discussion of the purpose of each of the five boxes. The text is mostly already in the manuscript, but should be organized around the figure itself. The figure has five boxes: I should see five paragraphs - or, more generally, five logical groupings of information.***

Please notice our response to your comment above (1.2). We have explicitly numbered the paragraphs that describe each step within the LUT configuration GUI (Figure 2) (see lines 155, 162, 169, 173 and 177 in the updated manuscript).

***1.5 (Section 3 – Proposal) Finally, the bottom half of your Figure 1 ("ALG - InternalFunctions") can be turned into it's own Figure 3. It should follow Figure 2 in the sense that you have to configure before you can run. Once again, five boxes means five paragraphs. Some of this text will be new and some can be derived from the existing section 3.***

We have followed the recommendation of the reviewer and included a Figure 4 (page 8) with focus on the ALG's internal functions for RTM model execution and LUT generation. The new figure includes numbered boxes whose content is further described in the paragraphs below. In order to facilitate readiness, we have added expressions such as "In step 1…", which serve to identify the paragraph with respect to the boxes in Figures 4 (see lines 193, 205, 210 and 212).

***1.6 (Additional comments) Lines 161-168… This discussion belongs somewhere else. You're telling the user that once you've built the LUT on a discrete grid, you can interpolate to any continuous value within that grid without losing much accuracy. That information should appear in the paper, but Section 3 really focuses how the model gets constructed (not used). I'd propose deferring this paragraph until you've completed your description of how the LUT was created in the first place.***

We thanks the suggestion but we don't fully agree on moving the Lines 161-168 in Section 3.2. In our opinion, these lines are not (in principle) related to the interpolation and how the LUT is used. In fact, we find that it is important to describe here the difference between "discrete" and "continuous" variables. Each of them has a slightly different interface to configure the variable values in the LUT, and so they are important to mention here as they are related with how to

use ALG. Notice that the name "discrete" refers to variables that can only take specific (integer) values (e.g. an aerosol model can only be rural, maritime or urban). The name "continuous" indicates that a variable can take any value (real numbers) (e.g., the solar zenith angle can be any value between 0º and 90º).

***1.7 (Additional comments) Lines 198-200 …: this discussion belongs somewhere else. The second set of lines discusses details of how the interpolation occurs. If you aggregate these two blocks, you end up with a useful description of how ALG gets used in real-world situations***

Thanks for the suggestion. However, we consider that the description of the interpolation functions fit well within the description of the LUT node distribution. Moving this text somewhere else would imply having a new section (3.4) with only a few lines explaining the interpolation methods provided to users of ALG.

***1.8 (Additional comments) Lines 207-218 + Lines 230-232: This is an important paragraph that tells the user exactly what you store in your LUT.  Give motivation as to why these particular quantities.***

This is indeed a good observation. We have rephrased the introductory sentence (see lines 215-216) to give further motivation on the interest of having the LUT defined as atmospheric transfer functions.

***1.9 (Additional comments) Also, you list several quantities in the LUT as bullet points, then switchgears to discuss a technical issue that differs between codes, and then back to completing LUT quantities, this time as numbered items.  Why not complete the bulleted list first and then follow up with the difficult LUT entry.***

We have followed the suggestion by the reviewer and moved the lines 230-232 (in the original paper line numbering) right after the list of atmospheric transfer functions (now in lines 230-233) in order to complete the description of the LUT content.

***1.10 (Additional comments) Finally, this is the spot where the authors have the opportunity to tell the reader the hard work that has clearly been done to make different radiative transfer codes, with different types of outputs, fill the exact same LUT.***

We have added a sentence to stress that the harmonization of RTM outputs is one of the key aspects of the constructions of LUTs in ALG (see lines 234-235).

***1.11 (Additional comments) Lines 219-230:  The text is fine, but please lead in with a motivating sentence. Something like "Most values stored in the ALG LUT can be obtained directly from standard RTM outputs. The exception is TOA radiance: obtaining this value differs depending on the RTM being used***

In fact, most RTM provide the TOA radiance spectrum as an output. The lines here were meant to support the idea that the atmospheric transfer functions can be used in forward and backward (atmospheric correction) simulations. Once said that, we agree that these lines should be reorganized in the text. We have decided to introduce equation 1 (now in line 220) before listing the content of the LUT in order to motivate why the atmospheric transfer functions are used to create the LUT.  Equation 2 must come after the explanation on the complexity to harmonize the different RTM outputs, since in 6SV the atmospheric transfer functions have slightly a different meaning (due to the uncoupling of absorption and scattering in 6SV) (see lines 238-239 and equation 2 in line 242).

We believe that the reorganization of the text results in a more clear structure to follow the understanding of ALG.

***1.12 (Additional comments) Figure 4: Please define "SI" somewhere (in the figure or in the text)***

Thanks for pointing to this missing acronym. SI (Sensitivity Index) is now included in the text (see line 257).

**Response from the Authors to** the comments from the **Anonymous Referee #2**:

*Dear anonymous referee*

*First, we would like to thank you for your interest in reviewing our publication. That being said, we clearly see the strengths of our work and tend to disagree with some of your remarks. Please allow us to show our arguments and eventual discrepancies to your comments. We hope that, through iteration, we can improve the quality of the paper.*

*Kind regards,*

*The authors*

**2.1 The paper is a technical software […] description and that the results of the model intercomparison […] are not very meaningful since the authors do not explain the reasons for discrepancies between the codes**

We acknowledge that this paper is a technical software description, as it is stated in the goals of our paper (introduction section at lines 48-49: "to describe the ALG tool from a functional and software design perspective") and in the conclusions section (line 322 "the main design concept and features of ALG have been described"). The intercomparison study is presented in this paper as an exercise (or application example) of how ALG could facilitate the tedious tasks of writing consistent input, running RTMs efficiently and harmonizing RTM outputs. That being said, we consider that the presented global sensitivity analysis results offer an additional comparative analysis of the studied RT models.

**2.2 The compared quantity (global sensitivity analysis) is not well suited for a model intercomparison, because it does not give much insight in reasons for discrepancies**

We consider that global sensitivity analysis can help to understand how the key input variables drive the spectral outputs. The global sensitivity analysis approach is able to quantify the sensitivity to each of the model variables and their interactions, which cannot be done using doing local analysis limited to a few wavelengths and geometric/atmospheric conditions (e.g. [Kotchenova et al. 2008]). Global sensitivity analysis provides therefore a statistical perspective of the entire (global) differences between models. Thus, these analyses provide a useful tool when used in combination to more specific/local analysis like [Kotchenova et al. 2008]. In fact, we also partly carried out the same analysis of Kotchenova et al. in the second part of Section 4 (lines 282 to 296). Our results are compatible with Kotchenova et al and demonstrates the consistency of the simulations. By no means we aim to replace the analysis done in previous validation papers, but to give a complementary analysis from the global sensitivity analysis perspective. Maybe the title of the paper is misleading. Accordingly, we have decided to change it to "Global sensitivity analysis comparison of …".

**2.3 providing exactly the same input is a difficult task, and obviously, the presented toolbox cannot generate exactly the same input**

We acknowledge the difficulty of this task. As you well indicate, RT models can have different description in their parameterization (e.g. aerosol optical properties, gaseous absorptions, vertical profiles or even definition of viewing/illumination geometric conditions). We are aware of that and that is why, in the last 3 years of ALG development, we have taken a special care of

ensuring consistency of model input configuration as much as the implemented RT models allow it. To do that, we have (1) studied carefully the RT user manuals, (2) done our internal analysis of input configuration and comparison of model outputs, and (3) contacted model developers (MODTRAN, libRadtran and 6SV) for user-support. The difficulty (or even impossibility) of ensuring exactly the same model input configuration using ALG is also not different than any other researcher would face when implementing their own RT model input generation, running and comparison scripts.

**2.4 I think that it is important that one works directly with the RT model instead of using a wrapper which hides the specific features of the individual models […] …the danger is that the users do not understand the physics behind radiative transfer, which they learn better when they work directly with the RT models**

We respect your opinion but tend to disagree. It is clear that users who want to have a deeper insight of the model will always work with the RT models directly if needed. Still, they can also benefit from the use of ALG e.g. to alleviate the complex tasks of model configuration and of execution of large datasets. Instead of being applied to ALG, this comment could be also applied to any other software tools designed for a specific RT model such as libRadtran's GUI, Ontar's PcModWin, MOSPAT, AEROgui and others (cited in the paper). However, nobody doubts about the usefulness of these tools to facilitate model configuration and execution. In that sense, ALG is not much different from these reference software tools, but goes beyond in offering a tool that harmonizes the RT model streamlining and parameter definitions (whenever possible). The toolbox does not hides the specific features of the individual models as the RTM input files written by ALG are always stored and available to users. Moreover, the tool is freely available to the community and users can participate as developers, having access to the (Matlab) code repository in GitLab. They can therefore report any eventual software bug or disagreement in input configuration and model outputs. We would like to add that the tool can potentially be used for education purposes (e.g. to master and PhD students in Remote Sensing). Under supervision of an expert/professor, students can learn the basics of atmosphere radiative transfer and get familiar with the inputs and outputs of RT models.

**2.5 I am a little skeptical whether the tool is really useful for the scientific community**

ALG v2.0 (and precursor unpublished versions) has been used in various scientific and industrial projects with related publications (see Section 5). Particularly, ALG is used in several ESA's FLEX mission activities such as end-to-end mission simulation (scene generation) and atmospheric correction. The toolbox is also used in a CNES project, in which atmospheric RT model comparison (MODTRAN6, libRadtran, 6SV, ARTDECO and SOSabs) and large database generation are key tasks. These examples shows the utility of ALG in practical scientific and industrial projects. Moreover, we have informally contacted and demonstrated ALG in front of RT model developers and users, and we have always received positive feedback and demonstration of interest in the tool. In addition, ALG is conceptually based on the ARTMO toolbox (www.artmotoolbox.com), which is a similar tool for the operation of a wide variety of vegetation (leaf-canopy) RT models. ARTMO is actively being used in several European scientific and industrial studies and several publications, conference presentations and tutorials demonstrate the interest of (and usefulness for) the scientific community. Our experience in ARTMO makes us confident that the tool can therefore be of interest for the remote sensing and atmospheric scientific community.

To conclude, we are confident that a broader scientific community might be interested in using the provided tool and that its use should not hamper the user understanding of RT models.

**Response from the Authors to** the comments from the **Executive Editor**:

***3.1 Currently the manuscript fails to comply with very significant parts of the GMD requirements for source code publication. GMD requires the public and persistent archiving of all of the source code on which a manuscript depends.***

We would like to thank you for informing us about the GMD policy on code and data availability. We realized that that indeed our paper was not correctly fulfilling the GMD policy as also indicated in the recent editorial note (https://doi.org/10.5194/gmd-12-2215-2019) and the full details on the requirements for data and code availability (https://www.geoscientific-model-development.net/about/code_and_data_policy.html)

We have carefully reviewed the information in these websites as well as the recommendations you kindly send us by personal e-mail. Accordingly, we have made available the source code of ALG v2.0 (used in this paper) as well as the generated data and scripts using the storage provided by Zenodo under the following link: https://doi.org/10.5281/zenodo.3555575. We have also updated the manuscript on the Code availability section (lines 340-346).

The source code is therefore freely available for the community through the Zenodo repository as well as in our own project website (www.artmotoolbox.com). We think that, with these modifications, the manuscript is now in line with the GMD policy on source code and data availability and thus that our manuscript is compliant for its publication in GMD.

[revised manuscript text omitted]
 L_{toa} = L_0 + \frac{(E_{dir}\mu_{il} + E_{dif})(T_{dir} + T_{dif})\rho}{\pi(1 - S\rho)} \tag{1}$$

where $\mu_{il}$ is the cosine of the SZA. The LUTs generated by ALG contain the atmospheric transfer functions used in Eq. (1) and which are described below:

  – The spectrum of intrinsically reflected radiance by the Earth's atmosphere ($L_0$ in $\mathrm{mW \cdot m^{-2} sr^{-1} nm^{-1}}$), also called atmospheric path radiance.

225  – The downwelling solar irradiance spectrum at surface level, splitted by its direct ($E_{dir}$) and diffuse ($E_{dif}$) fluxes, both in $\mathrm{mW \cdot m^{-2} nm^{-1}}$.

  – The atmospheric reflectance spectrum for the photons backscattered to the surface ($S$), also known as spherical albedo.

  – The upwelling direct and diffuse target-to-sensor transmittance spectra ($T_{dir}$ and $T_{dif}$).

In addition to these atmospheric transfer functions, the generated LUT file also includes:

230  – The extraterrestrial solar irradiance spectrum at 1AU Earth-to-Sun distance, $I_0$ in $\mathrm{mW \cdot m^{-2} nm^{-1}}$.

  – The wavelength vector at which these spectral magnitudes are calculated.

  – The name and values of the input atmospheric and geometric variables for each LUT node.

  – The values of the remaining (constant) parameters.

An important part of the complexity of ALG lies in being able to harmonize the different radiative transfer codes, with different types of outputs, to fill the exact same LUT. For MODTRAN simulations, these spectrally-dependent atmospheric transfer functions are automatically calculated by applying the interrogation technique presented in Guanter et al. (2009) and Verhoef and Bach (2012). In the case of libRadtran simulations, four runs are needed to compute these transfer functions (Debaecker et al., 2016). Similarly, 6SV directly provides the atmospheric transfer functions, however, with a slightly different definition due to the uncoupling of scattering and gas transmittance. The following transfer functions are used for 6SV: path radiance, at-surface total solar irradiance due to scattering ($E_{tot}$ in $\mathrm{mW \cdot m^{-2} nm^{-1}}$), total gas transmittance ($T_{gas}$), total upwelling transmittance due to scattering ($T_{tot}$) and spherical albedo ($S$). In this case, $L_{toa}$ is calculated through Eq. (2):

[revised manuscript text omitted]

---

## Author Response (AR2)

**Authors' response to Topical Editor**

Response from the Authors to the comments from the Topical Editor:

**Dear Dr. van Chiel**

First, thank you for your support reviewing our manuscript. We have taken your valuable comments into account and updated the manuscript accordingly. In particular, we have followed your advice on adding new plots with more measurable quantities. We have decided to plot the total solar irradiance directly and diffusely transmitted in the path Sun-target-sensor as well as the path radiance in order to better visualize the main radiative processes of absorption and scattering in the atmosphere for the different models. We believe that this has served us to improve the quality of the manuscript.

Regarding the additional minor comments, we have been taken them carefully into consideration. Please find in the text below the main changes of our paper, which are also highlighted in red in the updated manuscript to facilitate the revision process.

We trust that the comments raised through the revision process have been duly addressed to your satisfaction.

Kind regards,

The Authors

**Comment #1: Line 11. Intercomparison studies of what exactly?**

The utility of ALG is demonstrated in this manuscript through the comparison of top-ofatmosphere radiance simulations made with 3 broadly used radiative transfer models. The comparison is in particular carried out through global sensitivity analysis. We have rephrased this sentence in the Abstract for clarification.

**#2:** Section 2.3. The description of LibRadtran is a little minimal compared to the two above. We acknowledge your observation that this section is shorter when compared to the two previous sections (i.e., for MODTRAN and 6SV). We have added further description of the software package.

**#3: Figure 1 I do not fully understand why this figure is in the paper, since its content is literally in the text.**

We understand your comment and agree that the figure is of little use since its content is fully described in the text. Following the Editor recommendation, we have proceed to remove the figure from the manuscript.

**4: Section 4: In order to demonstrate your tool, in my view a demonstration of output in measurable quantities is indispensable. This also permits the user to validate the output of your tool against observations. A GSA is a complex and very useful quantity, but I cannot infer whether your tool produces the correct output based on its outcome. Please add some more simple plots to the paper in which for instance radiative fluxes are plotted against each other for some representative cases.**

**That being said, you do not have to solve the differences, but you should at least show that the developers of the respective RTMs have the opportunity to check the implementation in ALG.**

We appreciate the recommendation given by the Editor. We realize that indeed showing the spectral outputs generated by the execution of the different RTMs can be more illustrative to readers than the complex, though useful, GSA results. In the updated manuscript we have included the new Figure 4, which shows the path radiance, spherical albedo and total irradiance at TOA of MODTRAN, libRadtran and 6SV for a subset of the simulations carried out for GSA (see Table 1). Notice also the descriptive text around Figure 4 (in red), which gives further details about the observed differences between the analyzed models.

**#5: (Line 327)* Your paper contains only two plots. In my view, you can do better with your tool. Please show more plots that can be produced with your tool that can provide guidance on how to explain the differences.**

In response to the previous comment, we have added the new Figure 4, which shows the spectral output of the atmospheric LUTs as generated by ALG. This illustrative figure (three in fact) gives an idea of the harmonization of the various RTMs done automatically by the ALG tool. The figure is also giving further insight of some differences between the models. Other figures are possible with ALG (just to cite a few: spectral aerosol optical properties, vertical profiles of gas concentration and temperature, global maps of ECMWF variables), however these functionalities are not used for the global sensitivity analysis carried out here. Therefore, we prefer to limit the number of figures only to those that are relevant for the scientific content of the manuscript.

**#6 (Line 334) Is it possible to include Mystic solver (libRadtran) in ALG given the user license.**

The solver Mystic is implemented and freely distributed with the rest of the libRadtran package. We have already implemented the compatibility of ALG (v2.1) with Mystic and we are currently running simulations for several projects.

**#7 (Line 343)* What are the licenses of the underlying RTMs? Can you provide those freely with your tool? I am no expert in those things, but have you checked this thoroughly?**

We appreciate your remark. ALG does not re-distribute any third-party software (i.e. the atmospheric RTMs) to avoid problems with licenses. In fact, many licenses explicitly forbid the re-distribution (e.g. libRadtran or MODTRAN). ALG however provides guidelines (within its Help system) with instructions to download, compile and integrate these RTMs within ALG. We have added a short sentence in Line 375 describing this license aspects.